# Implementation of different HIV self-testing models with implications for HIV testing services during the COVID-19 pandemic: study protocol for secondary data analysis of the STAR Initiative in South Africa

Mohammed Majam,[1,2] Donaldson F Conserve [ID],[3] Vincent Zishiri,[1] Zelalem T Haile [ID],[4] Angela Tembo,[1] Jane Phiri,[1] Karin Hatzold,[5] Cheryl C Johnson,[6] Francois Venter[1,2]

MM, DFC and VZ are joint first authors.

For numbered affiliations see end of article.

**Correspondence to**
Mohammed Majam;
mmajam@wrhi.ac.za

## ABSTRACT

**Introduction** HIV self-testing (HIVST) presents a convenient, private approach that removes barriers to providing HIV testing services. The Self-Testing Africa (STAR) Initiative aims to scale up HIVST among priority and undertested populations. HIVST has the potential to help maintain testing services during the social distancing restrictions implemented to prevent the spread of COVID-19. This project evaluates linkage to confirmatory testing and treatment for HIV-positive clients for the STAR South Africa site.

**Methods and analysis** This secondary data analysis protocol aims to evaluate different HIVST distribution models from a prospective study implemented during November 2017 and December 2020 by Ezintsha, a subdivision of Wits Reproductive Health and HIV Institute. Routinely collected distribution and self-reported HIVST outcomes data will be deidentified and analysed. The main outcomes of interest are linkage to care and treatment among HIVST users who report a reactive HIVST result. Additionally, we plan to determine sociodemographic factors associated with linkage to care and treatment among HIVST users. Descriptive statistics will be used to describe the variables of interest, and modified Poisson regression with robust variance estimation will be performed to identify factors associated with linkage to care and treatment among HIVST users who report a reactive HIVST result. Risk ratios and 95% CIs for the risk ratios will be reported.

**Ethics and dissemination** The study protocol has been approved by the University of Witwatersrand Human Research Ethics Committee. The dissemination plan for the study findings will include presentations to local and international health authorities, international conferences and publications in open access journals.

---

## INTRODUCTION

An estimated 7.02 million South Africans are living with HIV, making the national

---

### Strengths and limitations of this study

► We investigate linkage to confirmatory testing and treatment for self-testers who report a positive self-test result.
► We will conduct secondary analysis of data from a prospective study that included different HIV self-testing distribution models.
► There is limited generalisability given that participation in self-reporting HIV self-test results is voluntary.
► To address this limitation, we followed up with as many participants as possible via the different self-reporting platforms.

---

population level HIV prevalence at 12.7% and 19.1% among those aged 15–49 years.[1] As part of the National Strategic Plan (NSP) on HIV, sexually transmitted infections and tuberculosis, 2017–2022, South Africa, has put in place various efforts to promote HIV testing services (HTS) in order to stop new HIV infections and to prevent AIDS-related deaths.[1] South Africa's 2017–2022 NSP sets a target to reduce the number of new HIV infections to under 100 000 by 2022.[1] . Routine HTS in South Africa largely takes a provider-based approach.[2] This approach requires that individuals present at an HIV testing location either at the health facility, in the community or in the home. While there have been gains in increasing access to HTS, an HIV testing gap remains largely among men and young people due to existing individual and socioeconomic barriers to accessing routine HTS among these populations. Closing this testing gap through increased access to HIV testing approaches that can provide HTS to

these undertested populations will be critical to achieve the Joint United Nations Programme on HIV/AIDS '95–95–95' goals.[3] However, there are new barriers to facility-based HTS in South Africa and elsewhere due to the implementation of quarantine, social distancing and other restrictions to prevent the spread of COVID-19.[4 5] HIV self-testing (HIVST) is a relatively new approach that provides an opportunity to reach, test, and diagnose or prevent infection among populations who were previously considered unreachable even during the COVID-19 pandemic due to the ability of people to self-test at home or in the facility while practising social distancing.[6–8]

The World Health Organization (WHO) recommends HIVST as an additional approach to providing HTS that could help with closing this testing gap by increasing access and acceptability for HIV testing.[2] HIVST presents a private and convenient and confidential approach to providing HTS that removes some of the barriers to routine HTS by allowing people to collect their sample and to receive their results in the privacy of their home without interacting with a healthcare professional.[6] HIVST also has the potential to reduce costs and to save time for the health delivery system and the end user by triaging out the negative patients.[6] HIVST kits are already available for purchase over the counter in several countries, including the USA, the UK, France, South Africa and Kenya, and have recently been introduced in the public sector in Malawi, Zambia and Zimbabwe.[9] Since December 2016, the WHO has provided guidelines to facilitate HIVST inclusion in national policies.[6] National HIV programmes worldwide have been requested to adapt policy and programme documents, including algorithms and training materials, to fully accommodate HIVST.[6] Consequently, the number of countries with supportive HIVST policies has grown rapidly to 77, with at least 38 countries currently implementing HIVST.[10] To support the use of HIVST in South Africa, guidelines for HIVST implementation were issued by the National Department of Health (NDOH) in February 2018.

### Self-Testing Africa (STAR) Initiative in South Africa

Despite the efforts to include HIVST in national policies, the market for HIVST was virtually non-existent in South Africa and other low-income and middle-income countries (LMICs) prior to the implementation of the HIV STAR Initiative.[11] Key challenges to the development of a healthy HIVST market notably included limited evidence on the public health impact and cost-effectiveness of HIVST, uncertain levels of consumer demand and concerns about potential social harms among others.[11] In response, Unitaid funded the STAR Initiative to establish evidence for HIVST safety, acceptability, feasibility and scalability. The STAR Initiative seeks to accelerate access to HIVST in LMICs by creating an enabling environment with regard to HIVST policies, generating diverse demand through multiple distribution channels adapted to the needs of priority populations and creating advocacy for additional financing, as well as accelerating market

entry for suppliers at affordable and sustainable prices.[11] The STAR Initiative has three phases, and the collaborators include the WHO, Population Services International and a consortium of partners, including London School of Hygiene and Tropical Medicine, Liverpool School of Tropical Medicine, University College London, Society for Family Health and the Wits Reproductive Health and HIV Institute (Wits RHI) in South Africa.[11] Phase I of the STAR Initiative was implemented in Malawi, Zambia and Zimbabwe (2015–2017) and demonstrated that HIVST can be used accurately, is widely accepted when offered through community, health facility-based and partner-delivered distribution models,[12 13] and importantly increases uptake of HIV testing among men, including among male partners of pregnant or postpartum women, adolescents, and vulnerable and key populations that do not otherwise use conventional testing services.[14] Building on these promising findings, STAR phase II was expanded to three new Southern African countries—South Africa, Swaziland and Lesotho—to introduce and inform HIVST scale-up[15 16] by establishing sustainable systems for HIVST delivery, linkage to confirmatory HIV testing, prevention and treatment services.[16] The new countries included in phase II represent three of the worst HIV epidemics in the world. STAR phase III, which was launched in 2020, included Cameroon, India, Indonesia, Mozambique, Nigeria, Tanzania, and Uganda to provide technical support for HIVST introduction and help pave the way for larger investments by the Global Fund for HIVST scale-up.

Wits RHI was one of the implementing partner of the STAR Initiative in South Africa and implemented the project in Gauteng, North West, Mpumalanga and Limpopo provinces of South Africa. Approximately 1.2 million HIVST kits were distributed, primarily targeting testing men, young people and key populations. Under the Wits RHI implemented programme, HIVST was offered if recipients did not know their current HIV status or received an HIV-negative result in the last HIV test taken >3 months ago. HIVST was not offered if the recipient was below the age of 12 or known HIV positive or if there was an identified risk of social harm, such as suicidal thoughts, gender-based violence or coerced testing.

This secondary data analysis protocol aims to evaluate different HIVST distribution models from a prospective HIVST study implemented in South Africa. The main outcomes of interest are linkage to care and treatment among HIVST users who reported a reactive HIVST result by examining the proportion of HIVST recipients who reported a reactive HIVST result and sought HIV confirmatory testing and entry into care among clients who received HIVST kits via the different distribution approaches implemented under the STAR Initiative in South Africa.

## METHODS AND ANALYSIS

### Study design

The study is a secondary analysis of routinely collected programme data from a prospective study aims at evaluating different HIVST distribution models implemented during November 2017 and December 2020 by Ezintsha, a subdivision of Wits RHI (see Data collection section). The different models implemented by Wits RHI are based on the NDOH HIV Self Screening Guidelines and are presented in table 1. Briefly, the models included community-based, public sector facility, key population and private sector targeted distribution. Each model had different channels and implementation strategies.

### Data collection

During HIVST distribution, recipients' age, gender, HIV testing frequency, as well as partner demographics were collected. Recipients were issued a barcode sticker that accompanies the HIVST kits. Duplicate barcodes were attached to the partner's HIVST kit(s) when provided. This barcode number was entered onto the participant's distribution data collection form and was used as the unique identifier for recipients and their associated HIVST kits issued. At distribution, HIVST recipients were also informed of the available options for support with testing, linkage to care as well as self-reported follow-up. Consent was sought from recipients for follow-up through one of the four self-reporting platforms offered on the project: telephonic, interactive voice response (IVR), Progressive WebApp (PWA) or WhatsApp messaging (business version). The different self-reporting platforms were offered to provide participants the ability to choose the option that may work best for them and inform which options will be included in the national HIVST policy to support the scale-up of HIVST and to help collect HIVST use and results data nationally. A standardised follow-up questionnaire was administered to consenting HIV self-testers and collected responses on the following: did you use the test? what was the HIV self-test result? did you request confirmatory testing? did you get confirmatory testing? were you linked to care? were you initiated on antiretorvial therapy (ART)? This set of questions was standard across the four follow-up platforms.

For HIVST recipients preferring follow-up by telephone, up to three telephone calls were conducted by a trained linkage officer at 2, 4 and 6 weeks post-distribution to obtain responses on the aforementioned described set of questions. IVR is an automated telephone system that interacts through short messages and calls with HIVST recipients to confidentially self-report their results. The first automated reminder SMS message to use and self-report HIVST results was sent within 24 hours following distribution. A second short message was sent on day 3 (SMS 2). If reporting had not been done at these two interactions, the IVR system automatically called the participant on day 7 to self-report results. Recipients also had the option to make an inbound call to the platform for support and self-reporting at any time postdistribution.

PWA is a mobile phone-based tool to support participants through self-testing and eventual confirmatory testing. It is accessible as a mobile site but is also downloadable to a phone, where the full site contents can be cached for offline browsing. Users that selected the web app received an SMS 24 hours following distribution, which provided them with a free URL link to access the PWA for registration on the application. Once subscribed, participants were able to access the website link and to self-report their results.

The Chatbot on a WhatsApp for business platform allowed users to send text and voice messages, make voice and video calls, and share images, documents, user locations and other media. HIVST recipients who selected follow-up and linkage support via the WhatsApp platform received a WhatsApp message 24 hours following distribution to confirm subscription. Once subscribed, participants were able to access the platform, which included terms and conditions, assistance through self-testing process and ultimately self-reporting of their results.

We will merge routinely collected programme distribution and HIVST users' self-reported data. The survey responses were captured and linked to HIVST recipients' distribution data via telephone number and barcode onto the project database. Data captured into project database were access controlled and regularly checked for quality, including data reconciliations and a data verification.

### Statistical analysis

Data for the secondary analysis will be deidentified and the analysis will be conducted between April and July 2021. Continuous variables will be expressed as means (±SD) or median and IQR as appropriate. Frequencies and proportions will be used to describe categorical variables. Binomial test will be used to determine whether the proportion of linkage to care and receipt of treatment among HIVST users equal to a prespecified proportion based on a recently published South Africa HIVST study.[17] Differences in the proportions of linkage to care and receipt of treatment among HIVST users by distribution modality and other sociodemographic characteristics will be assessed using $\chi^2$ test or Fisher's exact test, as appropriate. Unadjusted and multivariable adjusted relative risks of linkage to care and receipt of treatment among HIVST will be calculated using modified Poisson regression with robust variance estimation. Risk ratios and 95% CIs for the risk ratios will be reported. A two-tailed p value of <0.05 will be considered statistically significant. All analyses will be performed using STATA and SAS V.9.4.

### Handling missing data

Missing data will be handled using multiple imputation, which will ensure that as minimal as possible incomplete data are dropped. We will create numerous data sets containing different estimates of the missing values maintaining key identifying variables such as age, gender, testing frequency and distribution location, as well as

**Table 1** HIV self-testing distribution models implemented by Wits RHI/Ezintsha

| Model | Modality | Description |
|---|---|---|
| Community-based distribution | Hotspots | HIVST was offered to high-risk populations at hotspots through fixed pop-up sites. Hotspots include high foot traffic areas such as busy walkways or shopping centres. After a peer educator provided a demonstration on how to use HIVST kits, clients were provided with a choice to take a kit home or to use it onsite either on their own or with the assistance of the distributor. Clients opting to test onsite were given an opportunity to confirm reactive results. All women and men who indicated having a male partner were offered an HIVST test kit for their partners. 5% of consenting primary recipients of HIVST were telephonically followed up for outcomes of HIVST use, result, linkage to care and uptake of ART. From November 2019, all consenting clients were offered a choice to self-report their outcomes using one of the three mhealth platforms, specifically, WebApp, WhatsApp or on the interactive voice response system. |
| | Transport hub | The description of this modality is the same as the community-based hotspot modality, with a specific focus on distribution in transport hubs, including taxi ranks and train stations. |
| | Door-to-door | Within a specific mapped community or geographical location, peer educators and/or trained counsellors move from one household to another and offer HIVST to eligible clients residing there. Clients were offered HIVST with the option of either being directly assisted (in the presence of a healthcare provider) or unassisted. Clients who self-report reactive results were offered the option to confirm their results onsite. All women and men who had male partners were offered oral HIVST tests for their partners. To collect information on outcomes of HIVST use, result, linkage to care and uptake of ART, peer educators went back to the residences of all consenting clients. |
| | Integrated HTS (mobile) | This modality involved the integration of HIVST into existing mobile HIV testing activities. Clients were offered a choice between HIVST onsite or RDT with a counsellor. HIVST kits were distributed to individual clients after a demonstration of how to use them. Clients who opted for HIVST were only offered onsite testing with the option to confirm a reactive HIVST result immediately. Clients who confirmed reactive were referred to nearby health facilities for ART initiation. To collect information on outcomes of linkage to care and uptake of ART, linkage officers checked ART registers of referral health facilities. Integration of HIVST into mobile HTS did not displace traditional blood-based RDT uptake but provided a screening option of increased yield and efficiency for triaging clients for further managed healthcare. |
| Key population distribution | Sex worker | This modality was implemented alongside the Wits RHI sex worker programme, which provides various health services to sex workers at static sex worker clinics and at outreach sites. Sex workers were offered up to five HIVST kits to take for their network, which would either be their non-client sexual partners, their clients, other sex workers not accessing HTS or a member of the family or friend considered by the sex worker to be at high risk of HIV. This modality was intended to be a secondary only distribution modality; however, sex workers could use one kit for themselves if they needed to test themselves together with one of their recipients. All consenting sex workers were telephonically followed up to determine the usage, linkage to care and ART initiation status of their networks. |
| Public sector facility distribution | Outpatient department | Within the hospital outpatient waiting areas, peer educators created awareness and provided information on both traditional RDT and HIVST. Clients were then offered the choice between testing themselves onsite using either a blood or oral-based HIVST test inside a provided cubicle or having RDT with a counsellor. HIVST reactive clients were provided with the opportunity to confirm their results and to be initiated on ART immediately. Offering HIVST alongside traditional rapid testing in the outpatient departments provides clients with not only a choice but also a triage system whereby testers who screen non-reactive using HIVST may exit, allowing counsellors and nurses offering rapid testing enough time to focus on likely reactive testers. |
| Private sector facility distribution | GP practice and nurse-led clinics | Clients attending health services at GP practices or nurse-led Unjani clinics were offered HIV testing using HIVST during their consultation. After a demonstration was provided by the consulting clinician, clients were offered a private testing space within the facility to test themselves and were asked to self-report their result. Confirmatory testing and ART initiation were available onsite. All clients who self-report a reactive result were offered an HIVST test to take home for their partners. |

**Table 1** Continued

| Model | Modality | Description |
|-------|----------|-------------|
| Private sector distribution | Pharmacy | Pharmacists and pharmacist assistants at private pharmacies offered HIVST to adult clients, 17 and older seeking services that might suggest HIV risk, including emergency contraceptive, treatment for sexually transmitted infections, condoms, lubricants, sexual performance enhancers, etc. After a demonstration is offered, clients can choose to either test onsite on their own or take the kit home. Clients who chose to test onsite were given an opportunity to self-report their result to the pharmacist or the pharmacist assistant. All who self-report a reactive result were offered onsite confirmatory testing when available. All women and men who have male partners were offered oral HIVST tests for their partners. Clients were offered mHealth platforms to self-report on use, result and linkage to care. |
| | Workplace | This involved distribution of HIVST in formal workplaces with a focus on mining, manufacturing, construction, petroleum, agriculture and security sectors. Employees were offered an HIVST test kit to take home. All female employees were offered an HIVST kit for their male partners. 5% of the sample of primary consenting recipients of HIVST was telephonically followed up to obtain information on HIVST usage, result, linkage to care and ART initiation. Beginning August 2019, clients were also offered mHealth platforms to self-report on use, result and linkage to care. |
| Secondary-based distribution | Antenatal | After a demonstration and social harm assessment, all first visit antenatal attendees were offered HIVST to take home to support testing their sexual partners. The HIVST modality was not intended to replace facility based RDT testing but support testing of male partners. Women could take HIVST kit for herself to support home disclosure. Consenting women were followed up by telephone or at next visit to acquire information on whether HIVST had been used by partner(s) and for test result and linkage data. |
| | Index testing | HIVST is offered to newly diagnosed HIV-positive clients (ART clinic) to facilitate partner notification. Telephonic follow-up of HIV-positive index to ascertain index partner uptake of HIVST, linkage of positive sexual partner to confirmative testing and treatment initiation. |

GP, general practitioner; HIVSS, HIV self-screening; HTS, HIV testing service; mHealth, mobile health; RDT, rapid diagnostic testing; Wits RHI, Wits Reproductive Health and HIV Institute.

distribution model. Imputed data will be pooled into a study data set for analysis.

## Patient and public involvement

This study will not involve patients as it is a secondary analysis of routine programme data. HIVST recipients will not be involved in the development of research questions this study. However, findings from preference, usability and acceptability studies from STAR phase II will inform design of distribution approaches in the future . Wits RHI's community advisory commitees on adolescents, treatments, and prevention together with local community advisory boards provided ongoing guidance towards appropriate processes and procedures for implementation HIVST distribution and follow-up, particularly among adolescents and men for community-based distribution models. We plan to share the findings from this study through reports and presentations at local and international conferences. In addition, we will submit articles for publication to peer reviewed journals for wider visibility and use of the findings.

## Beneficiaries and target audiences

The study findings will contribute to the existing and growing body of evidence on the acceptability and efficacy of deploying HIVST and digital self-reporting and linkage to care platforms to reduce mortality and morbidity from HIV among men, key populations and adolescent boys and girls in South Africa. The project is of interest to a range of audiences, including academic and non-academic practitioners, community gate keepers for public health programmes targeting men and young people, HIV programme implementers, governments and public health funding organisations. A description of the primary and secondary outcomes of this study will help to identify distribution models for government and other funders to invest in scaling up of effective HIVST models. In addition, potential gaps in the implementation approaches and generalised lessons learnt can be drawn and inferred in future HIV programming for similar settings and target populations.

## Limitations

Our analysis plan could have limitations as follows. First, limited generalisability of the findings, given participation in self-reporting, is voluntary and not randomised. We recognise a high proportion of self-reporting a strong determinant of generalisability of the findings; therefore, we aimed to follow up as far as possible all consenting clients with as many as three reminders/engagement opportunities presented via the different self-reporting platforms. Second, we are prone to validity and reliability of secondary data, particularly where data are incomplete.

The project monitoring and evaluation team aims to ensure the completeness of data. In addition, analysis will be conducted on pooled imputed data sets to ensure that as minimal as possible incomplete data are dropped. Third, we acknowledge that both obtaining and reporting HIV test results are open to social desirability and non-response biases. The project provided self-reporting options to HIVST recipients and ensured anonymous reporting of HIVST use and results data. Fourth, while the analyses will provide results on the outcome for the different distribution modalities, the larger focus of the project and results are to provide preliminary evidence to create a supportive environment for the introduction and integration of HIVST in national policies, strategies, plans and regulations at the country level. Thus, the focus is less about comparison between different distribution modalities and more about piloting different implementation strategies to collect preliminary data to inform national policies, selection, adaptation, and scaling up of different HIVST delivery and linkage models for different populations.

## DISCUSSION

HIVST has been recommended by the WHO and several studies have demonstrated the its acceptability, feasibility and effectiveness.[2 6] However, there is limited evidence from implementation science research to provide guidance on how to scale up HIVST in LMICs for different hard to reach groups. The STAR Initiative aims to fill this gap, and the proposed secondary data will be a crucial step in generating the evidence needed to determine preliminary public health outcomes of HIVST in South Africa.[11] In addition, given the restrictions that COVID-19 has caused on HTS due to lockdown and social distancing requirements,[5 18] the findings of this study are crucial to inform the implementation and expansion of HIVST in order to sustain HIV testing in a safe and socially distant manner that can help reduce exposure to SARS-CoV-2. The different platforms for HIVST delivery evaluated in this study support the call for scaling up of HIVST during COVID-19 using a variety of strategies to ensure safe and appropriate delivery of HIVST and ongoing HTS.[7] Beyond the aforementioned implications, the findings will identify which distribution channels work best and should be adopted by the NDOH and included in the national HIVST policy as well as be incorporated in future WHO HIVST implementation guidelines to support national HIVST implementation plans.[16]

The Wits RHI STAR programme was implemented according to South Africa's national HIV self-screening guidelines. All clients voluntarily received the HIVST kits and consented to follow-up prior to the individuals' participation in the self-reporting surveys. Refusal of HIVST offer or consent to follow-up did not result in loss of access to any existing HTS. The follow-up surveys and secondary data analysis were approved by the University of Witwatersrand Human Research Ethics Committee

(ethics reference number 180405). The design and evaluation of the self-report platforms were approved by the University of Witwatersrand Human Research Ethics Committee ethics reference number 171113, and 180 708 for the IVR system, PWA and WhatsApp for business, respectively.

## ETHICS AND DISSEMINATION

The dissemination plan for the study findings will include presentations to local and international authorities at the NDOH, the WHO, and local and international conferences. In addition, the findings will be submitted for publications to open access journals with impact factor of >1.5 for wider visibility. To ensure that the findings help inform implementation of HIVST in areas most affected by the COVID-19 pandemic where HTS are disrupted, the findings will also be shared with community-based organisations and practitioners involved in HTS at the community level in case they do not access.

**Author affiliations**
[1]Ezintsha, a sub-division of Wits Reproductive Health and HIV Institute, University of Witwatersrand, Johannesburg, South Africa
[2]School of Clinical Medicine, Faculty of Health Sciences, University of the Witwatersrand, Johannesburg, Guateng, South Africa
[3]Department of Prevention and Community Health, Milken Institute School of Public Health, George Washington University, Washington, DC, USA
[4]Department of Social Medicine, Ohio University Heritage College of Osteopathic Medicine, Dublin, Ohio, USA
[5]Population Services International, Johannesburg, South Africa
[6]Global HIV, Hepatitis and STI Programmes, World Health Organization, Geneva, Switzerland

**Contributors** MM, DFC and VZ wrote the first draft of the manuscript. MM and FV designed the programme implementation strategies and the operational study design. MM, AT and VZ oversaw the implementation of the programme and drafted the standard operating procedures. JP oversaw data management processes. CCJ and KH provided an oversight of the programme implementation. DFC and ZTH helped revise the manuscript and address major comments from reviewers. All authors took part in the revision and reading and approved the final version of the protocol manuscript.

**Funding** This work was funded by the Unitaid (grant 2017–17-SFH-STAR) and the Bill & Melinda Gates Foundation (OPP1132929), the National Institute of Health (R00MH110343: awarded to DFC) and the HIV Dissemination Science Training Programme for Underrepresented Investigators (grant R25MH080665).

**Disclaimer** The findings and conclusions in this report are those of the authors and do not necessarily represent the views of the funder.

**Competing interests** None declared.

**Patient and public involvement** Patients and/or the public were not involved in the design, conduct, reporting or dissemination plans of this research.

**Patient consent for publication** Not required.

**Provenance and peer review** Not commissioned; externally peer reviewed.

**ORCID iDs**
Donaldson F Conserve http://orcid.org/0000-0001-8193-817X

Zelalem T Haile http://orcid.org/0000-0002-2912-8564

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
