## [Reviewer comments · BMJ Open]

ARTICLE DETAILS

TITLE (PROVISIONAL)	Implementation of Different HIV Self-Testing Models with Implications for HIV testing Services during the COVID-19 Pandemic: Study Protocol for Secondary Data Analysis of STAR South Africa Initiative
AUTHORS	Majam, Mohammed; Conserve, Donaldson; Zishiri, Vincent; Haile, Zelalem; Tembo, Angela; Phiri, Jane; Hatzold, Karin; Johnson, Cheryl; Venter, Francois

VERSION 1 – REVIEW

REVIEWER	McConnell, Margaret Harvard University T H Chan School of Public Health, Global Health and Population
REVIEW RETURNED	24-Jan-2021

GENERAL COMMENTS	Thank you for the revision of this manuscript. Further detail on the statistical analysis methods and COVID related context are improvements in the manuscript. However, it remains very unclear to me what the main outcome of the trial is. Specifically, how exactly is take-up defined? In the power calculation, the outcome seems to be proportion of the provided tests that are used. How will use of the test be measured -- by registering in the system? This needs to be clarified.
---

VERSION 1 – AUTHOR RESPONSE

Reviewer: Thank you for the revision of this manuscript. Further detail on the statistical analysis methods and COVID related context are improvements in the manuscript. However, it remains very unclear to me what the main outcome of the trial is. Specifically, how exactly is take-up defined? In the power calculation, the outcome seems to be proportion of the provided tests that are used. How will use of the test be measured -- by registering in the system? This needs to be clarified.

Response: We are happy that the revisions have helped to improve the manuscript and grateful for the additional question about the take-up outcome. We attempted to address this question in the previous revision. Given that our response was insufficient, we have decided to take your initial comment and recommendation (see below) that “take-up cannot be analyzed based on the data we have” and have taken out “take-up” as one of the outcomes and incorporated your initial recommendation by deleting “uptake of HIV self-testing” throughout the manuscript as an outcome.

Reviewer’s previous comment: However, given that the data will only report among those who agree to receive a self-testing kit, it does not appear that this version of take-up can be analyzed.

Response: The revised outcomes are below. We believe this will help to strengthen not only the analyses but also the contribution of the manuscript to the field because there are fewer papers that focus on examining linkage to care and treatment among self-testers.

The main outcomes of interest are linkage to care and treatment among HIVST users who reported a reactive HIVST result by examining proportion of HIVST recipients who reported a reactive HIVST result and sought HIV confirmatory testing and entry into care among clients who received HIVST kits via the different distribution approaches implemented under the STAR Initiative in South Africa.

VERSION 2 – REVIEW

REVIEWER	McConnell, Margaret Harvard University T H Chan School of Public Health, Global Health and Population
REVIEW RETURNED	12-Apr-2021
GENERAL COMMENTS	Thank you for your responses to my concerns about defining outcomes. The more limited set of outcomes is more appropriate to the planned study design.